



# The optical properties and in-situ observational evidence for the formation of brown carbon in cloud

Ziyong Guo[1,2,3], Yuxiang Yang[1,2,3], Xiaodong Hu[1,2,3], Xiaocong Peng[1,2,3], Yuzhen Fu[1,2,3], Wei Sun[1,2,3], Guohua Zhang[1,3,4], Duohong Chen[5], Xinhui Bi[1,3,4], Xinming Wang[1,3,4], Ping'an Peng[1,3,4]

[1] State Key Laboratory of Organic Geochemistry and Guangdong Key Laboratory of Environmental Protection and Resources Utilization, Guangzhou Institute of Geochemistry, Chinese Academy of Sciences, Guangzhou 510640, PR China

[2] University of Chinese Academy of Sciences, Beijing 100049, PR China

[3] CAS Center for Excellence in Deep Earth Science, Guangzhou, 510640, China

[4] Guangdong-Hong Kong-Macao Joint Laboratory for Environmental Pollution and Control, Guangzhou Institute of Geochemistry, Chinese Academy of Sciences, Guangzhou 510640, PR China

[5] State Environmental Protection Key Laboratory of Regional Air Quality Monitoring, Guangdong Environmental Monitoring Center, Guangzhou 510308, P. R. China

*Correspondence to*: Guohua Zhang (zhanggh@gig.ac.cn)

**Abstract.** Atmospheric brown carbon (BrC) makes a substantial contribution to aerosol light-absorbing and thus the global radiative forcing. Although BrC may change the lifetime of the cloud and ultimately affect precipitation, little is known regarding the optical properties and formation of BrC in the cloud. In the present study, the light-absorption properties of cloud droplet residual (cloud RES) were measured by coupled a ground-based counterflow virtual impactor (GCVI) and an Aethalometer (AE-33), in addition to the cloud interstitial (cloud INT) and ambient (cloud-free) particles by $PM_{2.5}$ inlet-AE-33, at Mt. Tianjing (1690 m a.s.l.), a remote mountain site in southern China, from November to December 2020. Meanwhile, the light-absorption and fluorescence properties of water-soluble organic carbon (WSOC) in the collected cloud water and $PM_{2.5}$ samples were also obtained, associated with the concentration of water-soluble ions. The mean light-absorption coefficient ($Abs_{370}$) of the cloud RES, cloud INT, and cloud-free particles were 0.25 ±0.15, 1.16 ±1.14, and 1.47 ±1.23 Mm⁻¹, respectively. The $Abs_{365}$ of WSOC was 0.11 ± 0.08 Mm⁻¹ in cloud water and 0.40 ± 0.31 Mm⁻¹ in $PM_{2.5}$, and the corresponding mass absorption efficiency ($MAE_{365}$) was 0.17 ±0.07 and 0.31 ±0.21 m² g⁻¹, respectively. A comparison of the light-absorption coefficient between BrC in the cloud RES/cloud INT and WSOC in cloud water/$PM_{2.5}$ indicates a considerable contribution (48-75%) of water-insoluble BrC to total BrC light-absorption. Secondary BrC estimated by minimum $R$ squared (MRS) method dominated the total BrC in cloud RES (67-85%), rather than in the cloud-free (11-16%) and cloud INT (9-23%) particles. It may indicate the formation of secondary BrC during cloud processing. Supporting evidence includes the enhanced WSOC and dominant contribution of secondary formation/biomass burning factor (> 80%) to $Abs_{365}$ in cloud water provided by Positive Matrix Factorization (PMF) analysis. In addition, we showed that the light-absorption of BrC in cloud water was closely related to humic-like substances and tyrosine/proteins-like substances (r > 0.63, $p < 0.01$), whereas only humic-like substances for $PM_{2.5}$, as identified by excitation-emission matrix fluorescence spectroscopy.



**Keywords**: brown carbon, light-absorption, cloud water, in-cloud process, cloud residuals



**Key points**

- The optical properties of BrC in both cloud-processed and cloud-free particles were simultaneously obtained.
- In-cloud process may facilitate the formation of BrC, with secondary BrC as the dominant fraction (67-85%).
- light-absorption of BrC in cloud water is closely related to humic-like and tyrosine/proteins-like substances.



## 1 Introduction

Brown carbon (BrC) makes a significant contribution to global radiative forcing, equivalent to 27-70% of that from black carbon (BC) (Lin et al., 2014a). The addition of BrC in climate models may change the direct radiation effect of organic aerosols from cooling (-0.08 W m$^{-2}$) to warming (+0.025 W m$^{-2}$), which may affect the lifetime and distribution of clouds, and thus precipitation and surface temperature (Zhuang et al., 2010). BrC may also contribute to uncertainties in global radiative forcing, as estimated to cause positive radiative forcing (-2.0 ~ +2.5 W m$^{-2}$, with an average of 0.01 ±0.04 W m$^{-2}$) in aerosol-

cloud interaction (Brown et al., 2018). However, such models rarely considered the secondary BrC, although increasing evidence shows that secondary BrC may represent the dominant fraction of total BrC (19-91%) (Wang et al., 2019a, 2019b). Aqueous-phase reactions in the cloud have been shown to significantly affect global secondary organic aerosol (SOA) production (Ervens, 2015; Liu et al., 2012; Spracklen et al., 2011), and thus may potentially contribute to secondary BrC. Production of BrC from aqueous-phase reactions have been extensively investigated in the laboratory, revealing that BrC can

also be secondarily formed through a variety of mechanisms, e.g., photochemical oxidation, nitration, and Maillard reactions (Lin et al., 2014b; Pósfai et al., 2004; Shapiro et al., 2009). For instance, secondary BrC is observed from the photo-oxidation of aromatics (Pang et al., 2019; Yang et al., 2021a), the nitration of phenol (Heal et al., 2007; Vione et al., 2001), and the reaction of carbonyls and ammonium/amines (De Haan et al., 2011; Nguyen et al., 2012; Heal et al., 2007). These light-absorption species such as nitrophenols, aromatic carbonyls, imidazole, and organosulfates have also been detected in

cloud/fog water (Desyaterik et al., 2013; Kim et al., 2019; Pratt et al., 2013; Bianco et al., 2016a; Lebedev et al., 2018; Lüttke and Levsen, 1997). However, to what extent do in-cloud processes contribute to the formation of BrC is still unclear.

Given that the currently applied imaginary refractive index of BrC based on the empirical formula of BC/OA ratio (Saleh et al., 2014) in the model simulation (Brown et al., 2018) may induce potential bias (Bikkina and Sarin, 2019), more field studies should be conducted to constrain the optical properties of BrC. Although various light-absorbing species have been identified

in cloud, only few studies focused on the optical properties of BrC in fog/cloud. Nitrophenols and aromatic carbonyls were the major fraction contributing to the light-absorption (~50%) of cloud water at wavelengths from 300 to 400 nm in Mt. Tai (Desyaterik et al., 2013). The mass absorption efficiency (MAE$_{365}$) of water-soluble organic carbon (WSOC) in fog water in California was 0.1-0.6 m$^2$ g$^{-1}$ (Kaur and Anastasio, 2017). The fluorescent chromophores of fog/cloud water, as identified by excitation-emission matrix fluorescence spectroscopy (EEMs) in Louisiana and Mt. Tai, were mainly composed of humic-like

and protein-like substances (Birdwell and Valsaraj, 2010; Zhao et al., 2019), which might also be related to the presence of BrC (Chen et al., 2016; Wang et al., 2020a). However, such studies were generally limited to PM$_{2.5}$, rather than in the cloud. Our previous studies have shown that the in-cloud aqueous-phase reactions could significantly promote the formation of SOA such as nitrogen-containing organic matters and affect the physicochemical properties of particles (Fu et al., 2020; Lian et al., 2021; Lin et al., 2017; Zhang et al., 2017a). In the present study, we took a further step to perform simultaneous on-line

measurements of the light-absorption coefficients for the cloud droplet residual (cloud RES), cloud interstitial (cloud INT), and ambient (cloud-free) particles, coupled with the light-absorption and concentration measurements of WSOC in cloud water





and PM$_{2.5}$. We aim to explore: 1) the optical properties of BrC in cloud-processed, cloud-free particles and WSOC in PM$_{2.5}$ and cloud water; 2) the possible contribution of in-cloud production to BrC light-absorption, and 3) the characteristics of fluorescent chromophores in cloud water and PM$_{2.5}$ and their relationship with light-absorption properties of BrC.

## 2 Methods

### 2.1 Sampling setup

Measurements of the cloud-free, cloud RES, and cloud INT particles were performed at Mt. Tianjing (24°41′56″N, 112°53′56″E, 1690 m a.s.l.) in Guangdong province, China during 18 November to 5 December 2020. This site is located at a national forest reserve and is less affected by anthropogenic sources. The cloud event determination threshold was set as visibility less than 3 km and relative humidity (RH) larger than 95%. During the cloud events, the cloud RES and cloud INT particles were alternately introduced into the instruments through ground-based counterflow virtual impactor (GCVI, model 1205, Brechtel Mfg., Inc., USA) and PM$_{2.5}$ cutoff, respectively, at a frequency of one hour. The GCVI cut size was set to 7.5 μm, where the transmission efficiency of cloud droplets is 50% (Shingler et al., 2012). The collected cloud droplets passed through an evaporation chamber (40°C), resulting in the cloud RES particles for downstream analysis. An Aethalometer (model AE-33, Magee Scientific., USA) was used to measure the light-absorption coefficients of particles at wavelengths of 370, 470, 520, 590, 660, 880, and 950 nm. AE-33 uses two parallel spot measurement technology to compensate for the light attenuation due to the filter loading effect (Drinovec et al., 2015). The BC concentration was calculated by the light-absorption coefficient at 880 nm. The detection limit of BC is less than 10 ng m$^{-3}$ (equal to 0.077 Mm$^{-1}$ at 880 nm) and the uncertainty is ~ 2 ng m$^{-3}$ (equal to 0.015 Mm$^{-1}$ at 880 nm), with a time resolution of 1 minute.

Cloud water samples were collected by a Caltech Active Strand Cloud water Collector, Version 2 (CASCC2) (Demoz et al., 1996; Yang et al., 2021b) when the visibility was less than 200 m (during 14 November to 4 December 2020). The collection efficiency was 50% at a cut size of 3.5 μm. During the sampling period, 53 cloud water samples were collected. The 0.22 μm quartz fiber filter was used immediately to remove insoluble components after collection of cloud water and then frozen at -20°C until analysis. Meanwhile, PM$_{2.5}$ samples were collected by a mid-volume (300 L min$^{-1}$) aerosol sampler (PM-PUF-300, Mingye, China). Daily samples (during 14 November to 8 December 2020) were collected on the quartz fiber filters, which were prebaked at 450°C for 4 h in a muffle furnace to remove residual organics before use. After collection, all samples were frozen at -20°C until analysis. In this study, PM$_{2.5}$ samples collected at the same time with cloud water samples were regarded as INT-PM$_{2.5}$ (n = 13), and the others as FREE-PM$_{2.5}$ (n = 19). It should be noted that some FREE-PM$_{2.5}$ samples also experienced short cloud events during collection. Blank samples of the cloud water and PM$_{2.5}$ were collected and processed following the same procedure as the samples.

### 2.2 Calculation of secondary BrC light-absorption



The light-absorption coefficient ($Abs_{BrC}(\lambda)$, Mm$^{-1}$) of BrC in different wavelengths can be obtained by AE-33, assuming that the absorption Ångström exponent (AAE) of BC is 1 and the light-absorption at 880 nm only due to BC (Drinovec et al., 2015).

The cloud RES, cloud INT, and cloud-free particles were generally located in submicron size (Fig. S1), and thus unlikely originated from non-combustion sources are mostly biogenic and mainly exist in the coarse mode (Perrino and Marcovecchio, 2016). The $Abs_{BrC}(\lambda)$ contributed by the combustion sources can be estimated through a BC-tracer method (Wu et al., 2018):

$$Abs_{pri,comb}(\lambda) = (\frac{Abs(\lambda)}{BC})_{pri} \times [BC]$$

Where Abs($\lambda$) is the total light-absorption coefficient of carbonaceous aerosol that measured by AE-33, $(\frac{Abs(\lambda)}{BC})_{pri}$ can be determined by the minimum $R$ squared (MRS) method to further evaluate the relative contribution of primary BrC and

secondary formation BrC to the overall $Abs_{BrC}(\lambda)$. Firstly, $Abs_{pri,comb}(\lambda)$ is calculated based on $(\frac{Abs(\lambda)}{BC})_{pri}$, which is assumed to be step increasing from 0 to 120 with a rate of 0.1. The target $(\frac{Abs(\lambda)}{BC})_{pri}$ value can be retrieved when the correlation coefficient ($R^2$) between $Abs_{BrC,sec}(\lambda)$ with BC concentration reaches the minimum (see Fig. S2). Previous studies showed that the bias of MRS method is less than 23%, when the measurement uncertainty is less than 20% (Wu and Yu, 2016). It should be noted that when the measured ratio of $\frac{Abs(\lambda)}{BC}$ is lower than the retrieved $(\frac{Abs(\lambda)}{BC})_{pri}$, the $Abs_{BrC,sec}(\lambda)$ could be negative. In

these cases, $Abs_{BrC,sec}(\lambda)$ is set to zero for subsequent analysis (Kaskaoutis et al., 2021; Wang et al., 2019a). These cases account for less than 5% in the cloud RES and 28-70% in the cloud INT and cloud-free particles

**2.3 Measurements of PM$_{2.5}$ and cloud water**

PM$_{2.5}$ samples were ultra-sonically extracted with ultrapure water (resistivity: 18.2 MΩ cm) for 30 min, then filtered by the 0.22 μm polytetrafluoroethylene (PTFE) filters to obtain the PM$_{2.5}$ aqueous extract. The concentrations of water-soluble ions,

water-soluble heavy metals, WSOC in PM$_{2.5}$ aqueous extract and cloud water samples were analyzed by ion chromatography (Metrohm 883 IC plus, Switzerland), inductively coupled plasma mass spectrometry (ICP-MS, Thermo Fisher, USA), and total organic carbon analyzer (TOC-V, Shimadzu, Japan), respectively. Parallel analyses showed that the relative standard deviation of each analysis was generally less than 15%. The reported concentration data herein was after blank subtraction.

The light-absorption coefficient ($Abs_{WSOC,\lambda}$) of WSOC can be obtained (Hecobian et al., 2010) by the measurement of cloud

water and PM$_{2.5}$ aqueous extract, with UV-Vis (UV1901, Kejie, China):

$$Abs_{WSOC,\lambda} = (A_\lambda - A_{700}) \times \frac{V_l}{V_a \times L} \times \ln(10)$$

Where $A_\lambda$ is the absorbance of the sample, $A_{700}$ is used to account for any drift; $V_l$ is the volume of ultrapure water used to extract the sample (for cloud water it is the total sample volume), $V_a$ is the volume of sampled air through the PTFE filter (for cloud water it is the total volume of sampled air), and $L$ is the cuvette path length (0.01 m).

The AAE values describing the spectral dependence of WSOC light-absorption can be further deduced by exponential fitting $Abs_{WSOC,\lambda}$ between 300-500 nm. The MAE$_{WSOC,\lambda}$ can be calculated by divided $Abs_{WSOC,\lambda}$ by the mass concentration of WSOC





(μg·m$^{-3}$). The E$_{250}$/E$_{365}$ (the ratio of absorbance at 250 nm to that at 365 nm) is used to describe the humification of organic matter, which is inversely related to aromaticity and molecular weight of WSOC (Kristensen et al., 2015). Specific UV absorbance (SUVA, m$^2$ g$^{-1}$,) at 254 and 280 nm had been proved to be qualitatively related to the structural characteristics
(aromaticity and molecular weight) of WSOC to a certain extent (Weishaar et al., 2003), which can be calculated using the following equation:

$$SUVA_{254\ or\ 280} = \frac{A}{L \times C_{WSOC}}$$

Where $A$ is the absorbance of sample at 254 or 280 nm, $C_{WSOC}$ is the concentration of WSOC (mg L$^{-1}$).

The excitation-emission matrix fluorescence spectroscopy (EEMs) of PM$_{2.5}$ extract and cloud water were measured by a
fluorescence spectrophotometer (F97pro, Lengguang, China). The sample blank was deducted before analysis, and the EEMs were normalized to the Raman units (R.U.) by using the Ramen peak (Ex = 350 nm, Em = 365-430 nm) of ultrapure water measured simultaneously with the sample (Lawaetz and Stedmon, 2009). Parallel factor (PARAFAC) analysis was performed on the acquired spectra with drEEM toolbox (version 0.3.0) based on MATLAB (Murphy et al., 2013), according to the outlier tests of PARAFAC, 6 samples with high leverage and high residual signals were removed in the modeling of PM$_{2.5}$ aqueous
extract. The details for obtaining maximum fluorescence intensity (F$_{max}$), fluorescence index (FI), recent autochthonous contribution (BIX), and humification index (HIX) were described in the supporting information (SI) text S1.

## 3 Results and Discussion

### 3.1 The optical properties of BrC during cloud events

The presence of BrC could be indicated by the AAE values derived from AE-33 data, which are 1.30 ±0.12 for cloud-free,
1.36 ±0.22 for cloud INT, and 1.32 ±0.15 for cloud RES particles. The light-absorption coefficient of BrC at 370 nm (Abs$_{370}$) of cloud-free, cloud INT and cloud RES particles are 1.47 ±1.23, 1.16 ±1.14, and 0.25 ±0.15 Mm$^{-1}$, respectively (Fig. 1), with the AAE values of BrC at 2.71 ±0.69, 3.13 ±0.97, and 2.76 ±0.89, respectively. The contribution of BrC light-absorption to the total particle light-absorption in the cloud-free, cloud INT, and cloud RES particles shows no significant difference, on average decreasing from ~23% at 370 nm to ~7% at 660 nm, as shown in Fig. 2.

For the cloud water and PM$_{2.5}$ aqueous extracts, light-absorption properties of WSOC at 365 nm are taken as the representative to those of water-soluble BrC (WS-BrC) in the present study. As expected, there is a positive correlation between Abs$_{365}$ and WSOC concentration in cloud water and PM$_{2.5}$ aqueous extracts (r > 0.61, p < 0.01). As shown in Fig. 1, there is great difference in Abs$_{365}$ of WSOC in FREE-PM$_{2.5}$, INT-PM$_{2.5}$, cloud water-Day, and cloud water-Night, which are 0.49 ±0.34, 0.27 ±0.18, 0.09 ±0.04, and 0.13 ±0.10 Mm$^{-1}$, respectively. The Abs$_{365}$ of WSOC in PM$_{2.5}$ in this study is at the same magnitude to that
of PM$_{10}$ in Tibet Plateau (Kirillova et al., 2016), and much lower than those in urban areas (as summarized in Table S1) (Chen et al., 2018, 2020; Huang et al., 2020; Kim et al., 2016). The AAE of WSOC has no significant difference among FREE-PM$_{2.5}$,





INT-PM$_{2.5}$, cloud water-Day, and cloud water-Night, which are 6.01 ± 0.81, 5.37 ± 1.08, 5.81 ± 1.47, and 6.31 ± 1.51, respectively, within the reported range.

The MAE$_{365}$ of WSOC in FREE-PM$_{2.5}$, INT-PM$_{2.5}$, cloud water-Day, and cloud water-Night are 0.31 ±0.17, 0.31 ±0.26, 0.17 ±0.07, and 0.17 ±0.07 m$^2$ g$^{-1}$, respectively. The MAE$_{365}$ of WSOC in cloud water and PM$_{2.5}$ are much lower than those in urban/alpine areas and various source emission samples (Table S1) (Chen et al., 2018, 2020; Fan et al., 2016; Huang et al., 2020; Kim et al., 2016; Kirillova et al., 2016; Li et al., 2019; Park and Yu, 2016; Soleimanian et al., 2020; Wu et al., 2019). The MAE$_{365}$ of WSOC shows no significant difference between the FREE-PM$_{2.5}$ and INT-PM$_{2.5}$, which is similar to the result observed in the Indo-Gangetic plain (Choudhary et al., 2018), but their values are quite higher, i.e., 1.6 m$^2$ g$^{-1}$ and 1.8 m$^2$ g$^{-1}$ for the INT-PM$_{1.0}$ and FREE-PM$_{1.0}$, respectively. The MAE$_{365}$ of WSOC in cloud water (0.06-0.32 m$^2$ g$^{-1}$) is slightly lower than the previously reported values in fog water (0.1-0.6 m$^2$ g$^{-1}$) in California (Kaur and Anastasio, 2017). Both the MAE$_{365}$ of WSOC in cloud water and PM$_{2.5}$ show a positive correlation (r > 0.84, $p < 0.01$) with SUVA$_{254/280}$, and a medium negative correlation (r > 0.43, $p < 0.05$) with E$_{250}$/E$_{365}$, which indicates that the aromaticity and molecular weight of WSOC may influence the light-absorption capacity of cloud water and PM$_{2.5}$ (Fig. S3).

Although there are tight correlations between the Abs$_{370}$ for cloud water and the cloud RES particles, and for the INT-PM$_{2.5}$ and the cloud INT particles (Fig. 3, r > 0.60, $p < 0.01$), the Abs$_{370}$ of WSOC in cloud water (0.12 Mm$^{-1}$) and INT-PM$_{2.5}$ (0.27 Mm$^{-1}$) is considerably lower than those in the cloud RES (0.24 Mm$^{-1}$) and cloud INT particles (1.08 Mm$^{-1}$) that collected simultaneously. Such difference may be attributed to the contribution of water-insoluble organic carbon (WIOC). The different optical properties for the whole BrC and WS-BrC may also be reflected by the AAE values. They are generally in a range of 4-8 at 300-500 nm in cloud water and PM$_{2.5}$, much higher than those for BrC (2-4) calculated from AE-33 data at 370-660 nm. The contribution of water-insoluble BrC to the light-absorption is estimated to be ~75% for the cloud INT particles and ~48% for the cloud RES particles on average, based on these differences (Fig. 3). It is also noted that the light-absorption of WIOC might still be underestimated by ~16% when sampling size is considered for the GCVI and cloud sampler (as discussed in SI text S1). High contributions of WIOC to BrC light-absorption have also been observed in the Indo-Gangetic plain (77%) (Satish et al., 2020), Beijing (62%), and Xi'an (51%) (Huang et al., 2020).

### 3.2 The secondary contribution of BrC during cloud events

Fig. 2 shows the contribution of secondary BrC to the total BrC in cloud-free, cloud INT, and cloud RES particles estimated by the MRS method. 11-16% and 9-23% of the total absorption of BrC come from the secondary BrC for the cloud-free and cloud INT particles, respectively. Only a slight difference observed for the cloud-free and cloud INT particles, indicating that cloud processing may have limited influence on the cloud INT particles. It is noted that even during the cloud-free periods, RH was generally higher than 70% (Fig. S1). The contribution of secondary BrC in cloud INT and cloud-free particles are in the low range of reported values (as summarized in Table S2) (Gao et al., 2022; Kaskaoutis et al., 2021; Lin et al., 2021; Wang et al., 2019a, 2019b, 2020b, 2021; Zhang et al., 2020, 2021; Zhu et al., 2021).



Differently, the contribution of secondary formed BrC to the total BrC light-absorption is 67-85% in the cloud RES particles, surprisingly higher than those in the cloud-free and cloud INT particles. Such a high contribution may suggest the critical role of cloud processing in the formation of BrC. Compared with the relative contributions for the cloud-free and cloud INT particles, the importance of such a process in cloud droplets remarkably overrides that in wet particles. The significance of secondary water-soluble BrC formation in cloud droplets may also be reflected by the significant correlation between the $Abs_{365}$ of cloud water and $PM_{2.5}$ aqueous extract with SNA (sulfate, nitrate, and ammonium) ($r > 0.77$, $p < 0.01$), and NOx ($r > 0.58$, $p < 0.01$), as shown in Fig. S4. The SNA and NOx concentrations are higher at night than the daytime (Fig. S5), consistent with higher $Abs_{365}$ of cloud water at night. NOx may enhance the formation of nitrogen-containing organics (Seinfeld and Pandis, 2016; Yang et al., 2021a), potentially contributing to the light-absorption of cloud water (Desyaterik et al., 2013). In-cloud aqueous processes leading to more CHON compounds in cloud water than below-cloud atmospheric particles has also been observed (Boone et al., 2015). In addition, a comparison between the WSOM (WSOM = WSOC*1.8) normalized by $K^+$ (as a primary source tracer) in cloud water than INT-$PM_{2.5}$ (Fig. S6) also clearly indicates the enhanced formation of WSOM in cloud water. It is consistent with that the light-absorption of WSOC contributed more to the cloud RES (~52%) than the cloud INT (25%) particles, as estimated in Fig. 3.

Consistently, the source and contribution apportionment of BrC in cloud water (i.e., $Abs_{365}$) evaluated by the PMF model (see SI for data analysis and evaluation methods) also supports the critical role of aqueous process on the formation of BrC, as shown in Fig. 4. Factor 1 is associated with relatively higher $K^+$, $NH_4^+$, $NO_3^-$, $SO_4^{2-}$, and $C_2O_4^{2-}$, contributing 64.3% to WSOC and 86.9% to $Abs_{365}$. It may be appropriately recognized as secondary products with contribution from biomass burning, as $K^+$ represents a tracer for biomass burning, and $NH_4^+$, $NO_3^-$, $C_2O_4^{2-}$, and $SO_4^{2-}$ are regarded as secondary species (Cheng et al., 2015; Wang et al., 2012). Note that $C_2O_4^{2-}$ is generally considered as a tracer of aqueous-phase processes (Zhang et al., 2017b). As previously observed, the aqueous SOA formed from biomass burning might contributed to the BrC budget in fog water (Gilardoni et al., 2016). Factor 2 is characterized by high levels of crustal trace elements such as $Mg^{2+}$, $Ca^{2+}$, Mn, and Zn, and thus identified as crustal materials, contributing 21.9% to WSOC and 8.7% to $Abs_{365}$. Factor 3 shows extremely high loading with $Na^+$ and relatively high $Mg^{2+}$, $Cl^-$, and Ni, which may originate from marine, contributing 13.8% to WSOC and 4.4% to $Abs_{365}$.

### 3.3 Fluorescence properties of BrC in $PM_{2.5}$ and cloud water

The results from the EEMs measurements further indicate the different characteristics of WSOC/WS-BrC in $PM_{2.5}$ and cloud water. Based on the PARAFAC model calculation (Fig. 5), two independent fluorescence components (P1-P2) assigned as humic-like substances are found in $PM_{2.5}$, whereas four independent fluorescence components (C1-C4) assigned as humic-like substances (C1-C3) and tyrosine/protein-like substances (C4) are found in cloud water (Catalá et al., 2015; Coble, 2007). The fluorescence components of cloud water are similar to those in Mt. Tai (Zhao et al., 2019) and France (Bianco et al., 2016b), where humic-like and protein-like substances are the main chromophores in cloud water. Compared with $PM_{2.5}$,



tyrosine/protein-like substances are unique to cloud water in the present study, which may be partly due to their relative enrichment in cloud water (Kristensson et al., 2010; Zhang and Anastasio, 2003).

In addition, the relative contribution of individual chromophores indicated by $F_{max}$ in $PM_{2.5}$ and cloud water also exhibits different characteristics, although humic-like substances are the dominant fluorescent fraction in both $PM_{2.5}$ and cloud water. The relative contribution shows no obvious difference between P1 and P2 components in $PM_{2.5}$ (FREE-$PM_{2.5}$ and INT-$PM_{2.5}$), whereas the C3 component contributes the largest (40.0%) to the fluorescent intensity in cloud water. Further analysis of the relationship between the fluorescent components ($F_{max}$) and the light-absorption of WSOC ($Abs_{365}$) in $PM_{2.5}$ and cloud water shows significant positive correlations between $F_{max}$ of all fluorescent components with $Abs_{365}$ ($r > 0.63$,$p < 0.01$, see Fig. 5). It suggests that these fluorescent components are tightly linked to the light-absorption of WSOC. The FI, BIX, and HIX of cloud water are $1.58 \pm 0.22$, $0.57 \pm 0.09$, and $4.99 \pm 3.83$ respectively, which indicates limited humified WSOC in cloud water, and also less affected by microorganisms and local sources (Huguet et al., 2009; McKnight et al., 2001; Zsolnay et al., 1999). Therefore, it is most likely that the organic components in cloud water may be significantly affected by in-cloud aqueous formation, consistent with the PMF results. With respect to the secondary processes, humic-like substances may be formed through Maillard reaction involving carbonyls and ammonium/amines (Bones et al., 2010; Hawkins et al., 2016), and also the photo-transformation of tyrosine (Berto et al., 2016).

**4 Conclusions and implications**

In the present study, the light-absorption properties of the cloud RES, cloud INT, and cloud-free particles were simultaneously investigated at a remote mountain site in southern China. Coupled with the measurements of light-absorption and fluorescence properties of WSOC in the collected cloud water and $PM_{2.5}$, it is evident that in-cloud aqueous processing facilitates the formation of BrC (i.e., 67-85% secondary BrC in cloud RES particles by MSR method). As potential contributors to light-absorption of BrC, only two fluorescence fractions of humic-like substances are found in $PM_{2.5}$, whereas four fluorescence fractions (three types of humic-like substances and one type of tyrosine/protein-like substances) are identified in cloud water, most likely attributed to secondary production. While extensive laboratory evidence indicated the possible formation of BrC in aqueous phase (Hems et al., 2021), our study represents the first attempt to show the possibility under real cloud condition. The results could support a previous hydrolysis that in-cloud formation of BrC might contribute to the enhanced BrC/BC in the attitude between 5-12 km (Zhang et al., 2017c). Such process might also have potential implication for the lifecycle of BrC (Liu et al., 2020).

In order to evaluate the influence of BrC formation in the light-absorption properties of cloud water, the imaginary part of the refractive index for cloud water was calculated according to Gelencsér et al. (2003), as detailed in the SI text S1. The average imaginary part of cloud water was $5.5 \times 10^{-8}$ at 365 nm (Fig. S7), ~10 times that of pure water. The imaginary part ($3.4 \times 10^{-8}$ at 475 nm) is a magnitude higher than previous laboratory simulation results ($5.2 \times 10^{-9}$ at 475 nm), involving 3,5-dihydroxy-benzoic acid reaction with $FeCl_3$ (Gelencsér et al., 2003). It should also be noted that it is the lowest estimation since only WSOC is included in the calculation. As previously indicated, the overall light-absorption of WIOC cannot be negligible.



According to the average $MAE_{550}$ and AAE of WSOC in cloud water and INT-$PM_{2.5}$, the optical properties of BrC during cloud events could be classified as weakly absorptive BrC (Saleh, 2020). The measured optical properties and suggested in-cloud formation of BrC would help better understand the atmospheric evolution and the radiation forcing of BrC.

*Supplement.* Supporting information includes one text (Text S1), seven figures (Fig. S1-S7), and two tables (Table S1-S3)
related to the manuscript.

*Data availability.* All the data can be obtained by contacting the corresponding author.

*Author contributions.* XB and GZ designed the research with input from XW and PP. ZG, XP, WS, HX and YY collected and
analysis samples. ZG processed data when XH and YY gave constructive discussion. ZG wrote the manuscript, and XB, GZ and YF interpreted data and edited the manuscript. XH, XP, and WS had an active role in supporting the sampling work. All authors contributed to the discussions of the results and refinement of the manuscript.

*Competing interests.* The authors declare that they have no conflict of interest.

*Financial support.* This work was funded by the Natural Science Foundation of Guangdong Province (2019B151502022), National Natural Science Foundation of China (42077322, 42130611, and 41877307), Youth Innovation Promotion Association CAS (2021354), and Guangdong Foundation for Program of Science and Technology Research (2019B121205006 and 2020B1212060053).



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





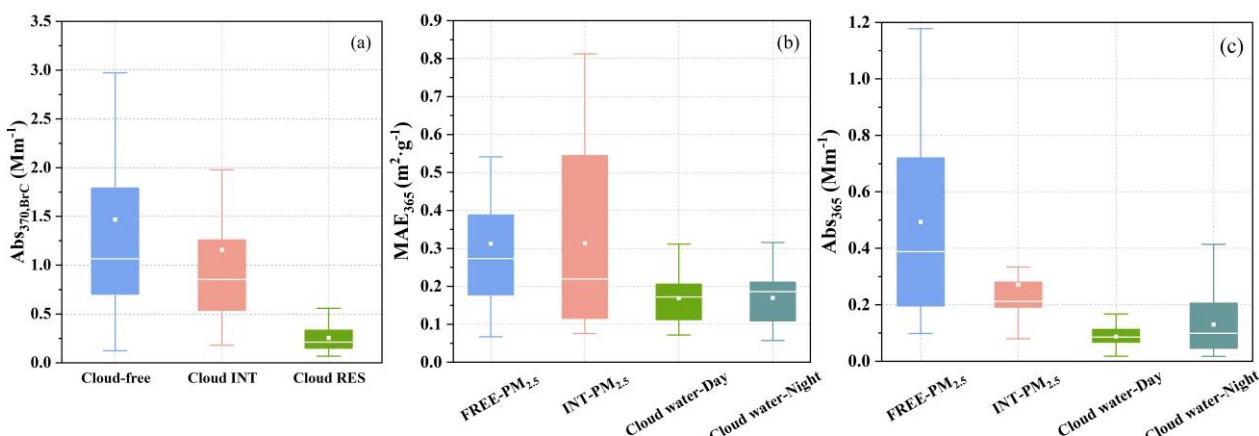

**Fig. 1. (a) The Abs$_{370}$ of cloud-free, cloud INT, and cloud RES particles, and (b) the MAE$_{365}$ and (c) Abs$_{365}$of FREE-PM$_{2.5}$, INT-PM$_{2.5}$, cloud water-Day, and cloud water-Night.**





**Fig. 2. The light-absorption of (a) BrC and BC; (b) primary BrC and secondary BrC at different wavelengths, and the percentage represent the contribution of (a) BrC light-absorption to the total particle light-absorption; (b) secondary BrC light-absorption to the total BrC light-absorption in the cloud-free, cloud INT, and cloud RES particles, respectively.**





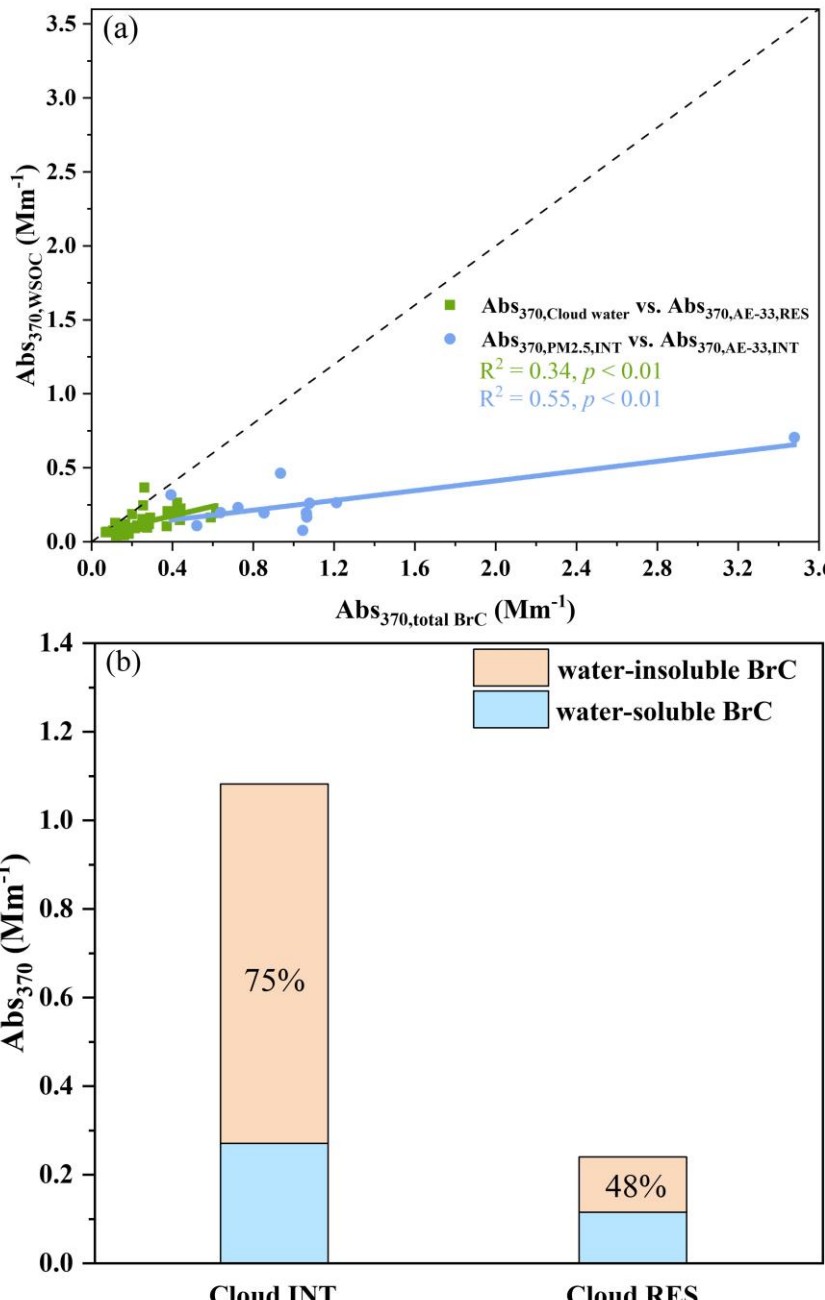

Fig. 3. (a) The correlations of WSOC light-absorption to total BrC light-absorption in 370 nm, and (b) the contribution of water-soluble BrC and water-insoluble BrC to total BrC light-absorption.





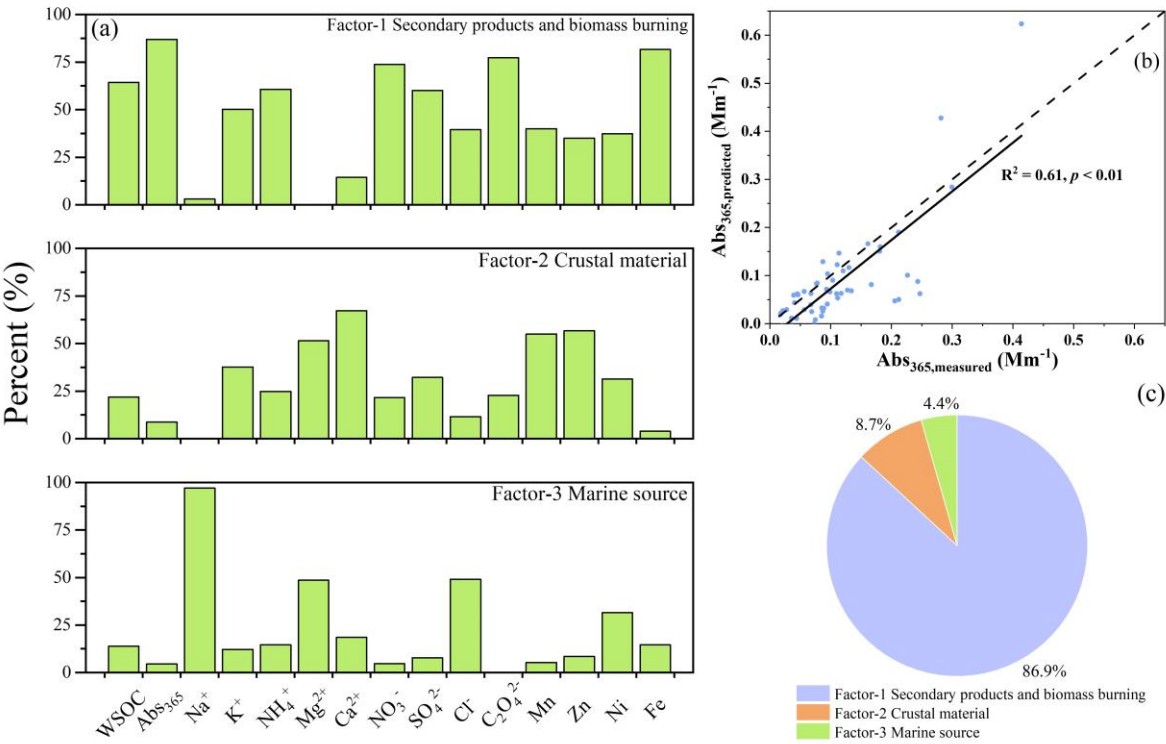

**Fig. 4. (a) The composition profiles (% of each species) for the three factors simulated of cloud water by PMF, and (b) the correlation of measured, and predicated Abs$_{365}$, and (c) the source apportionment for Abs$_{365}$ in cloud water.**



**Fig. 5. The EEMs components in PM$_{2.5}$ (P1-P2) and cloud water (C1-C4) that were identified by PARAFAC model, and the correlation between each chromophore F$_{max}$ and Abs$_{365}$ in (a) PM$_{2.5}$, and (b) cloud water.**