# Peer review of "The optical properties and in-situ observational evidence for the formation of brown carbon in cloud"

_Atmospheric Chemistry and Physics, 2021_

## Author Comment (AC1)

**Responses to comments by Referee 1**

General comments: This study attempted to investigate the role of cloud on the formation of brown carbon. A comprehensive and valuable dataset was collected, including the light-absorption properties of the cloud droplet residual, the cloud interstitial and cloud-free particles, the light-absorption and fluorescence properties of water-soluble organic carbon in the collected cloud water and PM2.5 samples, and the concentration of water-soluble ions. The presented data further indicate the formation of secondary BrC during cloud processing and a considerable contribution of water-insoluble BrC to total BrC light-absorption. Such results improve our understanding on the optical properties and secondary formation of BrC in cloud, and thus merit publication in ACP. Here are some minor issues that need to be addressed.

Reply: Thanks for the reviewer's positive comments.

Main comments:

Experiment section: why was PM2.5 inlet applied to rule out the cloud interstitial particles? Discussions should be provided on the possible uncertainty that may be introduced.

Reply: Thanks for the comments. It is assumed that the activated particles would grow to cloud droplets with median size at around 10 μm in the present study. This is reasonable since such a size distribution pattern have been previously observed at various regions such as at Mt. Tai (Li et al., 2017). In such case, the size of cloud droplets with size lower than 2.5 μm would be limited, and thus particles with size lower than 2.5 μm in cloud are regarded as cloud interstitial particles. We note that possible uncertainty could be introduced due to such an approximation. It would not lead to ambiguous conclusions since cloud residual particles were mainly focused.

The limitation has been added in the revised manuscript as "*It should be noted that the PM$_{2.5}$ inlet may introduce possible uncertainty for the collection of cloud interstitial particles due to the interference of cloud droplets, although the size distribution of cloud droplets were mainly concentrated on 6-9 µm at mountain sites (Li et al., 2017). However, it would not be the case when cloud residual particles were mainly focused in the present study.*"

**References**

Li, J., Wang, X., Chen, J., Zhu, C., Li, W., Li, C., Liu, L., Xu, C., Wen, L., Xue, L., Wang, W., Ding, A. and Herrmann, H.: Chemical composition and droplet size distribution of cloud at the summit of Mount Tai, China, Atmos. Chem. Phys., 17(16), 9885–9896, doi:10.5194/acp-17-9885-2017, 2017.

"The contribution of water-insoluble BrC to the light-absorption is estimated to be ~75% for the cloud INT particles and ~48% for the cloud RES particles on average, based on these differences (Fig. 3)." It is interesting to know that water-insoluble BrC contributes to such a high fraction of BrC in the cloud INT particles and the cloud RES particles. I wonder if some of this insoluble fraction is secondary origin.

Reply: Thanks for the comment. We further analyzed the correlation between the light-absorption of water-insoluble organic carbon (Abs$_{370,WIOC}$, as the difference of Abs$_{370, total\ BrC}$ and Abs$_{370,WSOC}$) with SNA concentration. There is a positive correlation for the cloud INT particles (r = 0.80, $p$ < 0.01) but no correlation for the cloud RES particles (r = -0.15, $p$ = 0.38), as shown in the figure below. This result might indicate possible secondary origin for the water-insoluble BrC in the cloud INT particles, yet there is no evidence for that in the cloud RES particles, which needs further investigations.

[Figure]

Figure 1 Correlation between $Abs_{370,WIOC}$ with SNA concentration in cloud RES and cloud INT particles

Lines 197: The authors presented correlation analysis between the Abs365 of cloud water and PM2.5 aqueous extract with SNA (sulfate, nitrate, and ammonium) (r > 0.77, p < 0.01), and NOx (r > 0.58, p < 0.01), and the result supports the secondary formation of BrC. Why was PM2.5 aqueous extract included in the analysis? Does this result also indicate the significance of secondary production of BrC in PM2.5?

Reply: Thanks for the comments. Correlation analysis between the $Abs_{365}$ of cloud water and $PM_{2.5}$ is compared to show if there are differences between cloud water and $PM_{2.5}$. The results also indicate possible secondary origin of BrC in the $PM_{2.5}$, which is similar to that observed for the cloud water. However, due to the limited size of samples (13 for INT-$PM_{2.5}$ and 19 for FREE-$PM_{2.5}$), the PMF method may introduce large uncertainty and thus cannot be used to estimate the secondary fraction of BrC.

Minor comments:

Line 53 what does "These light-absorption species" refer to?

Reply: The "*These light-absorption species*" refer to the products of the reaction that has been proved to produce secondary BrC in the laboratory. For more accurate expression, this sentence has been rewritten as "*The secondary BrC such as nitrophenols, aromatic carbonyls, imidazole, and organosulfates have also been detected in cloud/fog water*".

Line 134 "(SUVA, m2·g-1,)" error typo.

Reply: Thanks for the comment. The typo has been corrected to "*(SUVA, $m^2 \cdot g^{-1}$)*" in the revised manuscript

Line 156 "As expected, there is a positive correlation between Abs365 and WSOC concentration in cloud water and PM2.5 aqueous extracts (r > 0.61, p < 0.01)." Does it mean that WSOC in cloud water is mostly from PM2.5?

Reply: We are sorry for the misleading. Actually, this sentence means that $Abs_{365}$ poses a positive correlation with WSOC concentration both in the cloud water or $PM_{2.5}$ aqueous extracts. For accurate expression, this sentence has been revised to "*As expected, there is a positive correlation between $Abs_{365}$ and WSOC concentration in both cloud water and $PM_{2.5}$ aqueous extracts (r > 0.61, p < 0.01)*" in the revised manuscript.

Line 160 "much lower than those in urban areas (as summarized in Table S1)". I suggest to include the observed values.

Reply: Thanks for the reviewer's kind suggestion. The observed values have been added in the revised manuscript: "*much lower than those in urban areas (3.4-33.9 $Mm^{-1}$, as summarized in Table S1)*".

Line 197 what does "wet particles" refer to?

Reply: The "*wet particles*" refer to the cloud-free and cloud-INT particles. For better expression, the word "*wet particles*" has been replaced by "*cloud-free and cloud-INT particles*" in the revised manuscript.

Line 208 revise "Consistently, the source and contribution apportionment of BrC" to "the source apportionment of BrC".

Reply: Thanks for the reviewer's kind suggestion. We have replaced the sentence "*Consistently, the source and contribution apportionment of BrC*" with "*The source apportionment of BrC*" in the revised manuscript.

---

## Author Comment (AC2)

**Responses to comments by Referee 2**

General comments: This study presents the results of brown carbon measurements in cloud, including cloud droplet residuals, cloud interstitial particles, and cloud water. The authors attempted to demonstrate the role of cloud processing in the formation of brown carbon. The dataset covers both the collected cloud water and cloud residuals, and thus may offer new insight into cloud processing of brown carbon, which has been rarely investigated. The topic is appropriate for Atmospheric Chemistry and Physics, but there are some issues that need to be addressed before publication.

Reply: Thanks for the reviewer's positive comments.

Main comments:

Introduction: Generally, what are the major fractions contributing to the light-absorption of cloud water? The authors indicate that nitrophenols and aromatic carbonyls were the major fraction contributing to the light-absorption (~50%) of cloud water at Mt. Tai, but what about in other regions? Also, those related results for aerosol particles should be summarized herein.

Reply: Thanks for the reviewer's helpful comments. Although many light-absorption species such as nitrophenols, aromatic carbonyls, imidazole, and organosulfates have also been detected in cloud/fog water, there is only one research focusing on the optical properties of cloud water (Desyaterik et al., 2013), with the major light-absorption species detected as nitrophenols and aromatic carbonyls.

We agree with the comment, and the major light-absorption species of aerosol particles was also summarized in the revised manuscript: "*Many field studies focused on the optical properties of BrC in particulate matter. The light-absorption of BrC in $PM_{2.5}$ was well correlated with nitrophenols, polycyclic aromatic hydrocarbons, and oxygenated polycyclic aromatic (Wu et al., 2020). Nitrophenols and carbonyl oxygenated polycyclic aromatic hydrocarbons accounting 10-14% to the light-absorption at 365 nm in urban $PM_{2.5}$ (Huang et al., 2020). The contribution of nitrophenols and nitrated salicylic acids*

*to the aqueous extract light-absorption of PM₁₀ was 0.10-3.71% and 5 times higher than their mass contribution to WSOC (Teich et al., 2017).*"

**References**

Desyaterik, Y., Sun, Y., Shen, X., Lee, T., Wang, X., Wang, T. and Collett, J. L.: Speciation of "brown" carbon in cloud water impacted by agricultural biomass burning in eastern China, J. Geophys. Res. Atmos., 118(13), 7389–7399, doi:10.1002/jgrd.50561, 2013.

Huang, R. J., Yang, L., Shen, J., Yuan, W., Gong, Y., Guo, J., Cao, W., Duan, J., Ni, H., Zhu, C., Dai, W., Li, Y., Chen, Y., Chen, Q., Wu, Y., Zhang, R., Dusek, U., O'Dowd, C. and Hoffmann, T.: Water-Insoluble Organics Dominate Brown Carbon in Wintertime Urban Aerosol of China: Chemical Characteristics and Optical Properties, Environ. Sci. Technol., 54(13), 7836–7847, doi:10.1021/acs.est.0c01149, 2020.

Teich, M., Van Pinxteren, D., Wang, M., Kecorius, S., Wang, Z., Müller, T., Močnik, G. and Herrmann, H.: Contributions of nitrated aromatic compounds to the light absorption of water-soluble and particulate brown carbon in different atmospheric environments in Germany and China, Atmos. Chem. Phys., 17(3), 1653–1672, doi:10.5194/acp-17-1653-2017, 2017.

Wu, C., Wang, G., Li, J., Li, J., Cao, C., Ge, S., Xie, Y., Chen, J., Li, X., Xue, G., Wang, X., Zhao, Z. and Cao, F.: The characteristics of atmospheric brown carbon in Xi'an, inland China: Sources, size distributions and optical properties, Atmos. Chem. Phys., 20(4), 2017–2030, doi:10.5194/acp-20-2017-2020, 2020.

Section 3.1 Line 172 The discussions related to the influence of aromaticity and molecular weight of WSOC in the light-absorption capacity should be improved. What is the real meaning for a medium negative correlation ($r > 0.43$, $p < 0.05$) with E250/E365? Is such evidence consistent with those obtained by the EEMs measurements in section 3.2?

Reply: Thanks for the comments. The $E_{250}/E_{365}$ could be used as a qualitative measure of aromaticity and molecule weight, in which a lower $E_{250}/E_{365}$ ratio means higher aromaticity and larger molecule weight (Peuravuori and Pihlaja, 1997; Kristensen et al., 2015). The medium negative correlation between $MAE_{365}$ and $E_{250}/E_{365}$ indicates the limited influence of aromatic and molecule weight on the $MAE_{365}$. For more accurate expression, this sentence has been rewritten as "*Both the $MAE_{365}$ of WSOC in cloud water and $PM_{2.5}$ show a positive correlation ($r > 0.84$, $p < 0.01$) with $SUVA_{254/280}$, and a medium negative correlation ($r > 0.43$, $p < 0.05$) with $E_{250}/E_{365}$, which may indicate that higher $MAE_{365}$ of WSOC has higher aromatic and molecule weight, the aromaticity and molecular weight of WSOC may influence the light-absorption capacity of cloud water and $PM_{2.5}$*".

No meaningful results could be obtained through correlation analysis between the $E_{250}/E_{365}$ and fluorescent components ($F_{max}$). The $E_{250}/E_{365}$ ratio was used to analyze the possible influencing factors of the light-absorption of WSOC, where EEMs was used to investigate the possible chemical components of WSOC, the purposes and results of the two analyses are different.

**References**

Kristensen, T. B., Du, L., Nguyen, Q. T., Nøjgaard, J. K., Bender Koch, C., Faurskov Nielsen, O., Hallar, A. G., Lowenthal, D. H., Nekat, B., Van Pinxteren, D., Herrmann, H., Glasius, M., Kjaergaard, H. G. and Bilde, M.: Chemical properties of HULIS from three different environments, J. Atmos. Chem., doi:10.1007/s10874-015-9302-8, 2015.

Peuravuori, J. and Pihlaja, K.: Molecular size distribution and spectroscopic properties of aquatic humic substances, Anal. Chim. Acta, 337(2), 133–149, doi:10.1016/S0003-2670(96)00412-6, 1997.

Section 3.2: The authors came to the conclusion that NOx may enhance the formation of nitrogen-containing organics, based on the correlation analysis. I suggest including a discussion on the detailed mechanisms related to such a conclusion. Also, is there any evidence to exclude other pathways as indicated in the introduction?

Reply: Thanks for the reviewer's suggestion. As discussed in section 3.3, the humic-like substances may be formed through Maillard reaction involving carbonyls and ammonium/amines, however, we cannot exclude the other pathways such as photochemical oxidation mentioned in the introduction. The specific mechanism for the conclusion that NOx may enhance the formation of nitrogen-containing organics can be obtained from previous research, which was summarized in the revised manuscript: "*$NO_2^-$ resulted from the dissolved NOx can react with benzene and finally formed nitrophenol in the presence of UV-A (Harrison et al., 2005; Vione et al., 2004). Various of reactive oxygen/nitrogen species generated from the photolysis of inorganic nitrate in aqueous-phase could also facilitate the photooxidation of organic compounds to form BrC (Seinfeld and Pandis, 2016; Yang et al., 2021)*"

**References**

Harrison, M. A. J., Barra, S., Borghesi, D., Vione, D., Arsene, C. and Iulian Olariu, R.: Nitrated phenols in the atmosphere: A review, Atmos. Environ., 39(2), 231–248, doi:10.1016/j.atmosenv.2004.09.044, 2005.

Seinfeld, J. H. and Pandis, S. N.: Atmospheric Chemistry and Physics: From Air Pollution to Climate Change, John Wiley & Sons., 2016.

Vione, D., Maurino, V., Minero, C., Lucchiari, M. and Pelizzetti, E.: Nitration and hydroxylation of benzene in the presence of nitrite/nitrous acid in aqueous solution, Chemosphere, 56(11), 1049–1059, doi:10.1016/j.chemosphere.2004.05.027, 2004.

Yang, J., Au, W. C., Law, H., Lam, C. H. and Nah, T.: Formation and evolution of brown carbon during aqueous-phase nitrate-mediated photooxidation of guaiacol and 5-nitroguaiacol, Atmos. Environ., doi:10.1016/j.atmosenv.2021.118401, 2021.

Section 3.2: PMF model indicates a possible influence of biomass burning on the formation of secondary brown carbon. It would be much better to include and compare with those found for aerosol particles. The paper needs to provide more discussion on this issue.

Reply: Thanks for the reviewer's comments. We agree with the reviewer's comments such an additional analysis would benefit the discussions. We tried to investigate the source apportionment of all particulate phase BrC (FREE-PM$_{2.5}$ and INT-PM$_{2.5}$, n=33) through PMF model with input parameters the same as the cloud water. Two to five factors were evaluated and the results are summarized in Table.1. No meaningful results could be obtained for such a limited sample number, and thus such results were not included in the discussion.

Table 1 Q values for PMF analysis with different number of factors.

| Num. of factors | $R^{2\#}$ for all input species | $R^2$ for WSOC | $R^2$ for Abs$_{365}$ | $Q_{robust}$* | $Q_{robust}/Q_{theory}$ | Bootstrap (100 runs) |
|---|---|---|---|---|---|---|
| 2 | 0.03-0.93 | 0.55 | 0.46 | 2674.1 | 0.73 | >77 |
| 3 | 0.02-0.99 | 0.59 | 0.38 | 1770.8 | 0.72 | >68 |
| 4 | 0.29-0.99 | 0.66 | 0.33 | 1156.9 | 0.91 | >93 |
| 5 | 0.26-0.99 | 0.67 | 0.90 | 822.6 | 0.88 | >47 |

$^{\#}R^2$ between the observed and predicted species